# Automatic Facial Landmark Detection for Neurosurgical Mixed Reality Applications in MRI and CT scans using Deep Learning

**Mathijs de Boer**[1]                                   M.deBoer-41@umcutrecht.nl

**Lambertus W. Bartels**[1]                              W.Bartels@umcutrecht.nl

**Pierre A.J.T Robe**[2]                                 P.Robe@umcutrecht.nl

**Tristan P.C. van Doormaal**[2,3]                       T.P.C.vanDoormaal@umcutrecht.nl

[1] *Image Sciences Institute, University Medical Center Utrecht, Utrecht, The Netherlands*

[2] *Department of Neurosurgery, University Medical Center Utrecht, Utrecht, The Netherlands*

[3] *Department of Neurosurgery, University Hospital Zürich, Zürich, Switzerland*

## Abstract

Mixed reality (MxR) has the potential to revolutionize the way neurosurgical interventions are performed. However, the use of MxR in the operating room (OR) introduces new challenges, such as the registration of preoperative images to the patient. This paper presents a deep learning method for automatic detection of facial landmarks in Magnetic Resonance Imaging (MRI) and Computed Tomography (CT) scans, which can be used for image-to-patient registration in MxR applications. The method achieves a mean error of $4.02(\pm 2.65)$ mm in $3.55(\pm 1.53)$ seconds on a CPU. Apart from the nasion, no statistically significant differences were found between the performance of the method between the CT and MRI scans.

**Keywords:** Deep Learning, Facial Landmark Detection, Geometry, Mixed Reality, Neurosurgery

## 1. Introduction

Mixed reality (MxR) has the potential to enhance neurosurgical interventions by providing additional spatial information. However, the use of MxR in the operating room (OR) also introduces new challenges. To effectively use MxR for neuronavigation, the preoperative images of the patient need to be registered to the patient.

Facial landmarks could be used to perform the image-to-patient registration. The process would involve identifying the location of the same landmarks on the patient and in the preoperative images. Because preoperative imaging may be in the form of Magnetic Resonance Imaging (MRI) or Computed Tomography (CT) scans, the automatic method should be able to predict the landmarks in scans from both modalities.

In this paper, we present a method for the automatic detection of facial landmarks in head CT and MRI scans using deep learning.

## 2. Methods

**Data** The skin in 262 scans (125 CT, 137 MRI) was automatically segmented using nnU-Net models Isensee et al. (2021); De Boer et al. (2024). The segmentations were converted to surface models and manually annotated with 7 facial landmarks. These landmarks

being the nasion, left and right medial and lateral canthus, and left and right pre-auricle. The landmarks do not shift significantly between the moment the scan is made and patient arrival in the OR, which is why they were chosen. Finally, the labeled meshes were split into training and testing sets, using an 80:20 split, respectively. At train time, the train set was further split into a training and validation set with an 80:20 split.

**Pipeline** Each mesh was reprojected to a 1024 by 512 pixel equirectangular representation using a variation of the method described by Torres et al. (2021). Our main change was the use of a spherical projection, instead of a cylindrical one, which allowed for a simpler approach to back-projecting the predicted 2D coordinates to 3D coordinates and does not require setting vertical sampling bounds. The landmarks were also projected to 2D coordinates on the spherical map.

We used a neural network, based on the one by Kerfoot et al. (2021), to predict 2D coordinates of the landmarks. For training this network, the mean absolute error was modified to measure the angular error of the predicted and ground truth 2D coordinates. Finally, predicted 2D coordinates were then converted back to 3D coordinates using the inverse of the spherical projection method.

**Evaluation** Euclidian distances between the predicted and ground truth 3D landmarks were measured. The errors were not normally distributed, so a Mann-Whitney U test was used to compare the errors between the CT and MRI scans. A Šidák corrected p-value of 0.0073 was used as the significance threshold.

To assess feasibility of the method in time-critical situations, the time it took to fully process each mesh was measured. All predictions were performed with an Intel Core i9-9970X CPU. No GPU was used for predictions, as the time required to transfer the model and data to GPU memory was more than the additional time it would take to predict on the CPU.

## 3. Results

The mean Euclidian distances for all predicted and ground truth landmarks was $4.02(\pm2.65)$ mm. The nasion error showed a statistically signifance between MR and CT scans, no other statistically significant differences were found, see Figure 1. The mean time to predict was $3.55(\pm1.53)$ seconds.

## 4. Discussion

Our results show that the proposed method is able to quickly predict the facial landmarks with a clinically acceptable error. The method is also able to predict the landmarks accurately in both CT and MRI scans.

The nasion showed a significant difference between the CT and MRI scans, albeit not a strong one. It is possible that this is due to intra-observer variability and familiarity bias, as the CT scans were labeled first. Figure 1(b) shows that the nasion displays a larger vertical error distribution, which supports this theory. The method is also able to predict the landmarks in a short amount of time, which is important for its potential use in emergency situations.

While the method shows promise, further refinement and integration into a larger pipeline for neuronavigation are needed. Finally, the effects of inaccuracies on the overall registration accuracy need to be evaluated.

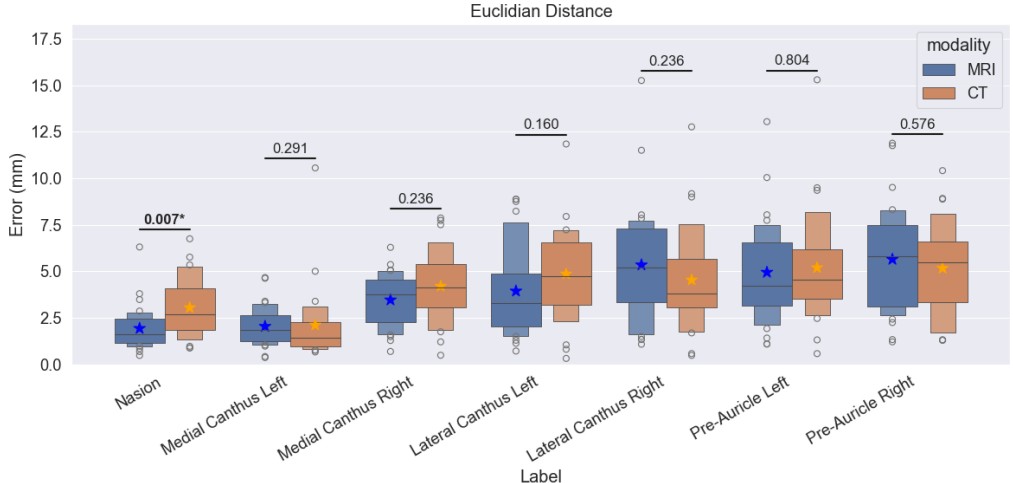

(a) Euclidian distances. Each landmark is marked with a star for the mean value, and a p-value for the Mann-Whitney U test between the MRI and CT values. Significance is marked with bold text and an asterisk.

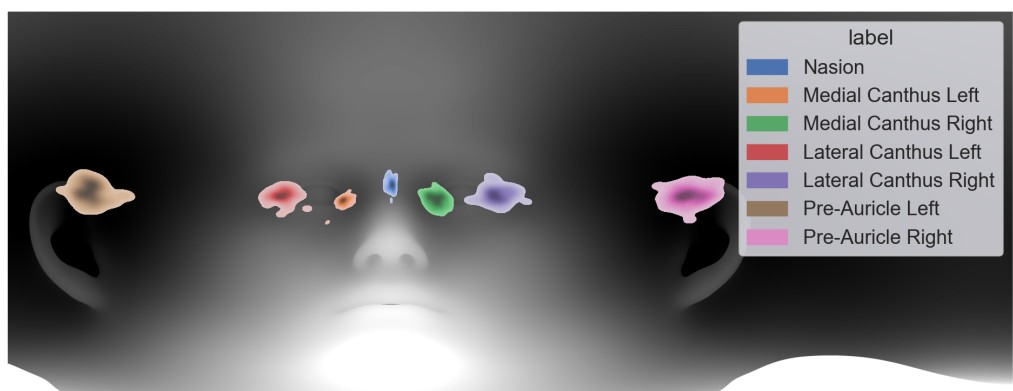

(b) Distribution of errors. The density of the distribution is visualized using desaturation.

Figure 1: Plots showing the results of the method.

## Acknowledgments

We would like to thank Jesse van Doormaal and Tessa Kos for their inputs during the development of the method. We would also like to thank the Hanarth Foundation for providing funding for the MISTICAL project. Finally, we thank The Blender Foundation for releasing the skin mesh we used for visualization under a CC-0 License.

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
