# OpenReview forum: "Automatic Facial Landmark Detection for Neurosurgical Mixed Reality Applications in MRI and CT scans using Deep Learning"
_MIDL.io/2024/Short_Papers — MIDL 2024 Short Papers_

### Official Review · Reviewer_QiXi · 2024-04-25

**Confidence:** 4
**Final Rating:** 4

**Review:**

This paper proposes to use deep learning to automatically detect facial landmarks in MRI and CT scans to register preoperative images to the patient for mixed reality in the OR.

Pros:
- Interesting CAI application of deep learning
- Methods and experimental settings are clear described
- Statistical analysis on results was performed
- Provided context that level of error and time for inference was clinically acceptable

Cons:
-  I am not an expert in mixed reality application in CAI, but I feel that while the paper claims that the level of error is acceptable, there is quite a large spread, and based on the visualization in 1b, I wonder if the larger errors would really be acceptable and if it propagates with other errors if it would be acceptable
- There is no information about where the scan data comes from

---

### Decision · Program_Chairs · 2024-04-26

Accept